# Feasibility of Using a Risk Assessment Tool to Predict Hospital Transfers or Death for Older People in Australian Residential Aged Care. A Retrospective Cohort Study

**DOI:** 10.3390/healthcare8030284

**Published:** 2020-08-21

**Authors:** Meidelynn Ooi, Ebony T Lewis, Julianne Brisbane, Evalynne Tubb, Tom McClean, Hassan Assareh, Ken Hillman, Helen Achat, Magnolia Cardona

**Affiliations:** 1Medical School, The University of New South Wales, Kensington 2052, Australia; meidelynnningwei.ooi@student.unsw.edu.au; 2School of Population Health, Faculty of Medicine, University of New South Wales, Kensington 2052, Australia; ebony.lewis@unsw.edu.au; 3School of Psychology, Faculty of Science, University of New South Wales, Kensington 2052, Australia; 4Uniting (Aged Care Services), Sydney 2067, Australia; jbrisbane@uniting.org (J.B.); etubb@uniting.org (E.T.); tmcclean@uniting.org (T.M.); 5Agency for Clinical Innovation, St Leonards 2065, Australia; hassan.assareh@health.nsw.gov.au; 6Intensive Care Unit, Liverpool Hospital, Liverpool 2170, Australia; k.hillman@unsw.edu.au; 7Western Sydney Local Health District, North Parramatta 2151, Australia; helen.achat@health.nsw.gov.au; 8Institute for Evidence-Based Healthcare, Bond University, Gold Coast 4226, Australia; 9EBP Professorial Unit, Gold Coast University Hospital, Southport 4215, Australia

**Keywords:** residential aged care, risk factors, advance care planning, hospital transfers, end of life

## Abstract

Residents of Aged Care Facilities (RACF) experience burdensome hospital transfers in the last year of life, which may lead to aggressive and potentially inappropriate hospital treatments. Anticipating these transfers by identifying risk factors could encourage end-of-life discussions that may change decisions to transfer. The aim was to examine the feasibility of identifying an end-of-life risk profile among RACF residents using a predictive tool to better anticipate predictors of hospital transfers, death or poor composite outcome of hospitalisation and/or death after initial assessment. A retrospective cohort study of 373 permanent residents aged 65+ years was conducted using objective clinical factors from records in nine RACFs in metropolitan Sydney, Australia. In total, 26.8% died and 34.3% experienced a composite outcome. Cox proportional hazard regression models confirmed the feasibility of estimating the level of risk for death or a poor composite outcome. Knowing this should provide opportunities to initiate advance care planning in RACFs, facilitating decision making near the end of life. We conclude that the current structure of electronic RACF databases could be enhanced to enable comprehensive assessment of the risk of hospital re-attendance without admission. Automation tools to facilitate the risk score calculation may encourage the adoption of prediction checklists and evaluation of their association with hospital transfers.

## 1. Introduction

The proportion of people aged 65 and over is expected to double by 2050 [1]. Their high prevalence of chronic comorbidities has implications for the current and future demand for residential aged care facilities (RACFs) [2] and frequent hospital transfers [3,4]. In Australia, RACF residents represent 0.2–2.4% of all emergency department (ED) transfers [3,5] with nearly 10% of hospital admissions in older adults being RACF residents [1] and 15% of presentations from the RACFs deemed preventable [6].

Over half of RACF residents die within a year of admission and many have not discussed their goals of care through advance care planning before a health crisis arises [7,8]. Hospital transfers for residents who are nearing the end of life (EOL) can be distressing and are often associated with burdensome medical interventions such as unsuccessful attempts to resuscitate patients [9]. These non-beneficial treatments result from delays in discussions about palliative goals of care as disease progresses, clinicians’ hesitation on prognosticating time to death [10] and from unrealistic societal expectations of survival. Yet, it is known that some of these hospital transfers could have been avoided through educational interventions that empower RACF nursing staff to effectively manage medical conditions on site [11]. A set of criteria has been developed for screening to identify older patients at risk of death within 3–4 months of assessment: the Criteria for Screening and Triaging to Appropriate aLternative care (CriSTAL) tool [12]. This is a checklist of objective parameters that can add risk points as a simple score to indicate which patients would benefit from an open discussion about end-of-life care. These factors, readily available from the hospital medical records, have been validated as predictors of short-term risk of death in older people seeking ED care [12,13,14]. For the purpose of testing in residential aged care, we used a modified version (Appendix A) adding four variables shown by others to be associated with poor outcomes in residential aged care: polypharmacy [15,16] falls [17], in a 6-month period, pneumonia [18] in the 3 months prior to RACF admission, and nutritional vulnerability [19].

In this study, we hypothesised that such an objective tool could identify individuals in RACFs at high risk of death over the next 6–12 months to also enhance clinicians’ confidence in prognosticating time to death and risk of hospital transfers. We anticipated that this information could assist future Advance Care Planning (ACP) efforts in RACFs such as written resident’s preferences for type future medical intervention or their withdrawal, “Do Not Resuscitate” orders [20] and “Do Not Hospitalise” orders. The ultimate benefit of the analysis would be to contribute to the provision of quality of care that aligns with the residents’ preferences and reduces non-beneficial interventions [21].

### 1.1. Aim

We aimed to determine the feasibility of using the CriSTAL tool [12,13] for the identification of people at risk of death or hospital transfers who may benefit from end-of-life discussions.

### 1.2. Objectives

To examine the feasibility of identifying RACF residents who are in the last 6–12 months of life using modified CriSTAL tool criteria.To better understand the independent predictors of hospital transfers and death (after initial assessment).To examine predictors of a composite outcome of hospital transfer and/or death (after initial assessment).

### 1.3. Outcomes

The planned primary outcome was predictors of hospital transfers following initial assessment.

Secondary outcomes were predictors of death after RACF admission, and predictors of poor composite outcome (i.e., either hospital admission or death). Comparisons between those who experienced the outcome and those who did not (i.e., survived and were not admitted to hospital) were used to test the CriSTAL prediction of events.

## 2. Materials and Methods

This was a retrospective cohort of older residents from nine Australian residential aged care facilities selected from a single service provider network that had harmonised datasets and established a partnership with one of our universities. The sample size was not pre-specified but depended on the number of eligible people with complete relevant data during the study period.

### 2.1. Participants

Inclusion criteria: Residents aged 65 years and above who were permanent residents of the RACF for at least one month between June 2015 to July 2017 who had data available from clinical assessment that included at least frailty or chronic disease in the first month of the cohort, and with varying follow-up data until the end of the study, transfer to another facility or until death if it occurred earlier. The choice of study period was based on the most recent two years of data available in the participating facilities. Exclusion Criteria: Respite clients and permanent residents who did not have complete baseline data for at least frailty or chronic medical conditions during the specified index timeframe.

### 2.2. Data Collection

De-identified demographic and medical information was retrieved from routinely collected datasets to document the clinical risk factors for short-term death guided by the validated CriSTAL tool [12,13,14]. These database items were derived from several available sources within the facilities. For instance, ‘any dementia diagnosis’ was based on prior clinical notes from the geriatrician evaluation, hospital discharge diagnosis, screening with Psychogeriatric Assessment Scales locally undertaken [22], general practitioner’s review, and the reports from the community-based aged care assessment team who determined the required level of support before entering the RACF. Every doctor, organization or hospital may have used different screening scales. No imputation was used for missing data. If crucial items such as age, sex, or event date were missing, the full record was excluded from analysis. For all remaining records, there were no losses to follow-up. For all residents, risk factors mentioned or supporting documentation present in the record for a risk factor assessment was documented as a ‘yes’; if there was no mention of the risk factor and no assessment documentation for it, the risk factor was assumed to not to be present. Hospital admission was defined as a stay for at least one night in a hospital ward hospital but it excluded people sent home on the same date of presentation. Data extracted in comma-separated-values CSV format separately from the RACF source for hospitalisation were linked in-house and a purpose-specific dataset was built for outcome analysis. To prevent measurement, outcome reporting and knowledge biases, independent quality assurance for each individual record was undertaken by two members of the research team, with at least one having a clinical background. To prevent analysis bias, the expert opinion of two statisticians was sought.

An index date was selected for each eligible resident based on the earliest clinical assessment result available for them within the study period. A baseline risk score was calculated according to the CriSTAL criteria. The CriSTAL tool was originally designed with the 29 items mostly related to presence of irreversible chronic illness and use of residential aged care or hospitalisation [12].

### 2.3. Ethical Considerations

No direct contact with residents or surrogates occurred. The study was approved by the South Eastern Sydney Human Research Ethics Committee (17/330 (LNR/17/POWH/651), and patient consent was waived as this was a retrospective record review of participants from previous years who may no longer be in the RACF or may have died. The research was considered a feasibility study to inform future quality improvement initiatives.

### 2.4. Statistical Analysis

Descriptive statistics are presented for the baseline CriSTAL risk factor distribution among survivors and deceased participants, with unadjusted chi-square and hazard ratios to examine the relationship between the risk factors and the outcomes. To account for confounding effects and differing follow-up time for RACF residents, Cox proportional hazard regression models with backwards elimination were used. We set a cut-off point of *p* < 0.15 (instead of 0.05) to retain factors that enabled meaningful clinical interpretation beyond statistical significance. This was done to prevent eliminating a potentially important confounder too early in the modeling process [23]. All base models included age and gender on the grounds of biological plausibility. Explanatory variables considered included a history of chronic conditions (chronic heart failure, chronic obstructive lung disease, stroke, history of myocardial infarction), frailty as documented using the clinical frailty syndrome, cognitive impairment including dementia, nutritional vulnerability, and history of 2 falls within 6 months. Conditions from the CriSTAL checklist were to be excluded from Cox regression analysis if counts were less than 10 for each. This applied to cancer, chronic kidney disease and chronic liver disease in this population. Two other CriSTAL variables could not be documented, as during the course of data extraction, it became clear that a history of pneumonia in the 6 months before admission was not available; and polypharmacy had to be excluded due to the ambiguity of the total numbers of medications prescribed and the limited extent of medication information amenable to retrieval from the records at each facility. Descriptive statistics used SPSS (IBM version 25, Armonk, NY, USA) and all multivariate analyses were conducted using SAS v9.4 (SAS Institute Inc., Cary, NC, USA).

## 3. Results

### 3.1. Participant Profile

A total of 421 records were examined, and, after excluding those with repeat RACF admissions, 373 patients were eligible for analysis (Figure 1). The median length of follow-up was 160 days (IQR 43−411). One in five residents (22.8%) had a CriSTAL score ≥ 6 suggestive of eligibility for an advance care planning discussion, but only 43 residents (11.5%) had a formal advance care directive documented during the study period.

The participants’ profile indicates a predominantly female (59.5%), non-Australian born (58.4%), non-partnered (64.3% Single/Separated/Divorced or widowed) sample of old residents (88.2% aged 75+ years, median 84, IQR 79−89), largely affected by various levels of frailty (97%) as measured by a score of 5+ according to the Clinical Frailty Scale criteria [24]. Over a third suffered from at least one of the target chronic conditions, notably cognitive impairment and nutritional vulnerability (Table 1).

### 3.2. Poor Composite Outcome

Of the 373 residents in the study, 34.3% had a poor composite outcome with 26.8% residents dying within a median time of 248 days from initial assessment (IQR 84−423) and 19.0% having a hospital transfer within a median 72 days of initial assessment (IQR 34−128). CriSTAL scores ranged from 2–7 with a mean score statistically significantly higher (5.11 SD = 1.02) for those who had a poor outcome than for those who did not have a poor outcome (4.63 SD = 0.99; *p* < 0.001). The decedents were also older (86.2 years vs. 83.4 years; *p* = 0.018), but there was no statistically significant association between age and poor composite outcome.

Unadjusted associations generally showed that only a few variables such as COPD and having falls in the previous six months were statistically significant predictors of death or composite outcome. This significance held after adjustment for multiple potential confounders on the initial survival model (Table 2). The unadjusted significant association of frailty with composite outcome and death disappeared after adjustments. Cognitive impairment was only weakly but significantly associated with death both in unadjusted and fully adjusted models. Having a hospitalisation any time in the study period significantly predicted death in both unadjusted and fully adjusted analysis. However, male sex was the only variable statistically significantly associated with hospital admission (Table 2).

The final adjusted survival analysis model after backwards elimination (Table 3) revealed that ≥2 falls in the past 6 months and chronic obstructive pulmonary disease (COPD) more than doubled the hazard of composite outcome or death. Being a male was a significant risk factor for hospitalisation but not for composite outcome or death. Cognitive impairment and hospital admission remained significant predictors of death.

## 4. Discussion

To our knowledge, this is the only Australian study conducted to assess whether a risk-stratification tool can predict poor short-term outcomes in RACFs. Our study adds to current knowledge by highlighting the feasibility of documenting key risk factors in RACF databases, and the need for improved clinical profiles to flag residents whose management warrants an end-of-life discussion as a matter of priority. Four of the CriSTAL items proved significantly associated with poor outcomes. COPD, and a history of two falls in the past 6 months, predict death and composite outcome (i.e., having a hospital admission and/or death). Cognitive impairment and prior hospitalisation also predicted death in the RACF population. Advanced age did not reach statistical significance on any of the three outcomes, but being a male added to the risk of hospitalisation. More detailed documentation of polypharmacy falls and their severity, and frequency of sameday hospital transfers without admission could value-add to the estimation of risk and the characterisation of unnecessary and burdensome transfers.

While these selected risk factors do not emerge as surprise predictors, having evidence of the magnitude of their individual impact on the risk of poor outcomes could be a catalyst for identifying high-risk residents on initial RACF assessment to encourage discussions on end-of-life care preferences. Factors for hospital transfers from residential aged care can be complex and go beyond residents’ clinical profile [25]. Anticipating acute exacerbations can prevent detrimental and unnecessary hospital transfers, and negotiating limitations of treatment, which are often absent from documentation in this vulnerable group on arrival at hospital [8] can facilitate informed decision making. Prior evidence of the feasibility of reducing burdensome hospital transfers from RACFs [11,26] motivated our study to identify older at-risk people to also prevent hospital-based adverse events.

Further clinical details and flags in these information systems are warranted to monitor individual and group risk profiles and patient outcomes.

### 4.1. Comparison with Other Studies

Our findings are consistent with those of a recent retrospective record review in residential aged care where COPD was a significant risk factor for death [27], and with a previous prevalence study demonstrating that falls in residential aged care contribute substantially to ambulance transfers and hospitalisations (85% of fall injury admissions) in Australia [17]. The RACF databases in our study did not document the impact of single instances of falls or whether they required hospital admission or were an adverse event of polypharmacy [28], but one in four residents (26.4%) had a history of two or more falls within the past 6 months and this indicator was significantly associated with risk of poor outcome in our analysis. Male sex has also been found to be a predictor of hospitalisation in RACF residents in a previous review [29]. This may be explained by personal preference or other factors not measured in our study.

As 97% of our study subjects were frail, we had no opportunity to contrast frailty as a predictor of death for aged care residents, although we are aware others have found this association. Malnutrition has been shown to coexist with frailty and be associated with increased risk of morbidity and mortality in RACFs [19]. However, this association was not found in our study in either univariate or multivariate analysis perhaps due to a small sample size (13.4%) of malnourished residents in this study or due to the recognised fact that malnutrition screening tools used in the facilities tend to have a low sensitivity when used by non-dietitians [30]. Overall, however, the CriSTAL checklist had multiple objective risk factors to potentially identify individuals at high-risk of transfers. We know of no other prediction tool being used for this purpose, as other studies have attempted to measure the avoidability of hospitalisation [31] or appropriateness of transfers [32] rather than characterise the individuals who would benefit from an honest end-of-life discussion, based on evidence of their high risk.

The low prevalence of ACP documentation in our vulnerable study population was similar to another retrospective Australian study of RACF residents presenting to hospital EDs (13.3%) [33] but much lower than an earlier study (37%) [8] and highlights the need to normalise EOL discussions. Given the large proportion of residents born outside of Australia in our sample, it would seem appropriate to investigate the influence of culture on the implementation of complex advance care planning procedures in this setting.

### 4.2. Limitations

We acknowledge some practical and methodological weaknesses of this study. The contribution of polypharmacy, pneumonia in recent months, ED presentation without hospitalisation, and ICU admission could not be examined due to the nature and structure of recording in the RACF databases. The reasons for hospital transfers were not routinely documented and could have been unrelated to individual clinical risk, such as RACF policies and practices [34] that compel staff to order ambulances sometimes for care sensitive conditions. For other evidence-based risk factors examined in our study, the numbers were too small to achieve statistical significance (malignancy, chronic kidney disease, chronic liver disease), or prevalence was too high to enable contrasting of differentials between groups (nutritional vulnerability, frailty). This does not mean the association does not exist, as the direction of effect remained the same after adjustment, but it could not be proven statistically in this target population. Deaths in hospital and when residents were discharged to die at home were not always known or recorded, hence the prediction of death presented in this study may have been an underestimate. The methodological decision to assume that absence of evidence for a CriSTAL item was equivalent to a “no” result for that item was a default chosen from real-life data recording practice. The small number of variables achieving significance for risk of all outcomes in our models is likely an underestimate reflecting a small number of people with each of the conditions on the CriSTAL checklist. The use of predictive tools in other settings has been shown to decrease re-attendances to EDs by older people [35]. Unfortunately, the RACF databases in our study were found to lack information on multiple hospital attendances with same-day returns to the RACF, which precluded the valuable investigation of risk factors for avoidable transfers of ambulatory-care-sensitive conditions.

### 4.3. Implications of This Study

Our findings confirm that routine data from RACFs can feasibly be used to identify at-risk individuals and predict outcomes. Selected objective CriSTAL criteria extracted from the RACF record can indeed predict death and poor composite outcome. Identifying these significant predictive variables on admission to RACF may enhance prognostic confidence and guide RACF clinicians in initiating early discussions on end-of-life goals with the residents’ families. While implementation of ACP in RACFs can be challenging in light of uncertainty of dying trajectory prediction [36], on-site management and avoidance of burdensome hospital transfers may be the appropriate pathway to prevent hospital-related adverse events [34]. The generalisability of these results may not apply to RACFs in countries with dissimilar hospital transfer policies and staff scope of practice [37].

It was unfortunate that the electronic RACF data did not contain all the CriSTAL items for accurate predictive analysis. As RACFs are not technically health services but residential services, these databases were designed to collect items for administrative purposes rather than for clinical care. Incompleteness of clinical parameters requiring interpretation caveats, variable data entry quality, and holding of hospitalisation details in a separate database, generated higher than anticipated resource requirements in the preparation of routine data for research and evaluation purposes. Lack of access to data on polypharmacy as per our definition and the low prevalence of malnutrition on our study mean that these findings should be interpreted with caution. An upgrade of RACF databases to enable automated risk score calculation, the inclusion of ED presentations with and without admission, and the enabling of further clinical documentation may suit the evaluation needs in the future. Following these modifications, a larger scale prospective study would more appropriately investigate the feasibility of the CriSTAL tool to evaluate the accuracy of prediction of short-term mortality and hospital transfers. Future research could also study whether knowledge of the predicted risk triggers advance care planning or the choice to manage residents on site.

## 5. Conclusions

Our adjusted results from this retrospective cohort analysis indicate that the presence of COPD and history of falls are independent significant predictors that can flag residential aged care individuals at high risk of events such as composite poor outcome and death. Cognitive impairment and hospitalisation further raise flags for risk of death, while male sex adds to the risk of hospitalisation. Knowing this for individual residents can improve their end of life by triggering advance care planning discussions on admission to the RACF. While it is feasible to identify other multiple predictors of death from RACF records, the variable prevalence of these factors and absence of RACF information about the extent of sameday hospital transfers precluded accurate quantification of one of our objectives. Incomplete off-site mortality ascertainment is also amenable to improvement. Given the known high predictive ability of the CriSTAL parameters in hospital settings, we would recommend the enhancement of the electronic systems in residential aged care to enable risk stratification, group analysis, linkage with hospital transfers, and inclusion of comprehensive risk parameters in a future upgrade of RACF databases so the collection can also serve clinical planning purposes.

## Figures and Tables

**Figure 1 healthcare-08-00284-f001:**
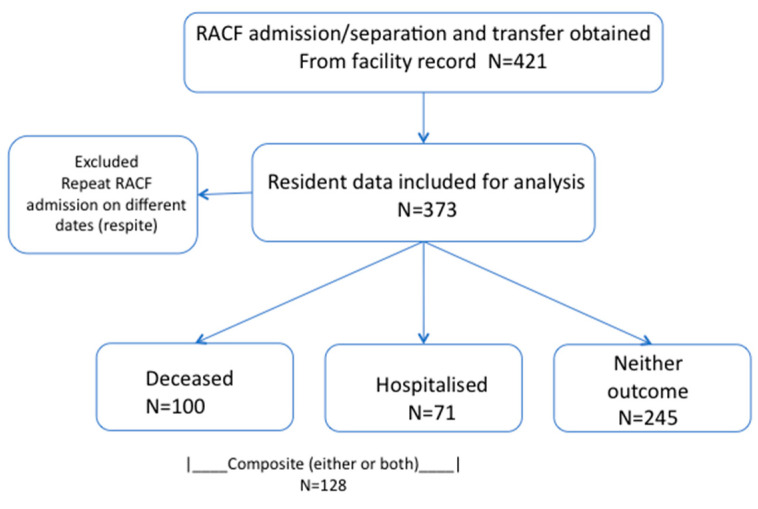
Resident eligibility, exclusion and relevant outcomes.

**Table 1 healthcare-08-00284-t001:** Clinical characteristics of participants in 9 residential aged care facilities.

Variable	*n* = 373	Percentage (%)
Frailty		
Not Frail	11	2.9
Mild/Moderately Frail	181	48.5
Severely Frail	181	48.5
Chronic Conditions		
Advanced Malignancy ǂ	7	1.9
Chronic Kidney Disease	3	0.8
Congestive Heart Failure	15	4.0
Chronic Obstructive Pulmonary Disease	41	11.0
Cerebrovascular Disease	64	17.2
Myocardial Infarction	14	3.8
Liver Disease	1	0.3
≥1 chronic condition	129	34.6
≥2 chronic conditions	16	4.3
Cognitive Impairment		
Dementia	217	58.2
Long Term Mental illness (Anxiety/Depression)	141	37.8
Behavioural Alterations	7	1.9
≥1 Cognitive Impairment ψ	277	74.3
Nutritional Vulnerability		
Sarcopenia	1	0.3
Malnutrition ∞	50	13.4
Unintentional Weight Loss (>3 kg)	20	5.4
Feeding Tube	3	0.8
Feeding Dependency Ω	22	5.9
Modified Diet Types	84	22.5
≥1 Nutritional Vulnerability Criteria §	141	37.8
≥2 Falls in 6 Months	98	26.4

ǂ Stage 3 or 4 cancer”, “advanced cancers” “malignant tumours” or “metastasis”, “malignancy”. ψ Presence of ≥ 1: dementia, long-term mental disorder, behavioural alterations, mental disability from stroke. ∞ MNA score of 7 or below indicates malnutrition. Ω Not independent on feeding, needing manual or other feeding assistance. § Presence of sarcopenia, malnutrition, unintentional weight loss (>5 kg), feeding tube, and modified diet types.

**Table 2 healthcare-08-00284-t002:** Unadjusted and adjusted (full initial model) analysis for predictors of poor composite outcome, and single outcome of death or hospital admission.

	Unadjusted Poor Composite Outcomea	Fully Adjusted Poor Composite Outcome ^a^	Unadjusted Death Outcome	Fully Adjusted Death Outcome	Unadjusted Hospital Admission	Fully Adjusted Hospital Admission
	HR (95% CI)	HR (95% CI)	HR (95% CI)	HR (95% CI)	HR (95% CI)	HR (95% CI)
Age (continuous)	1.02 (0.99–1.04)	1.01 (0.99–1.04)	1.03 (1.00–1.06) *	1.03 (0.99–1.06)	1.00 (0.97–1.04)	1.00 (0.97–1.04)
Male sex	1.34 (0.94–1.90)	1.43 (0.98–2.07)	1.23 (0.823–1.83)	1.12 (0.72–1.74)	1.68 (1.05–2.70) *	1.72 (1.05–2.82) *
Severe Frailty (as per CFS) ^b^	1.45 (1.02–2.07) *	1.26 (0.86–1.84)	1.58 (1.05–2.38) *	1.26 (0.81–1.96)	1.26 (0.792–2.02)	1.25 (0.75-)2.08
Chronic heart failure	1.46 (0.68–3.14)	1.61 (0.73–3.55)	1.62 (0.71–3.72)	1.31 (0.54–3.17)	1.81 (0.73–4.52)	2.06 (0.79–5.36)
COPD ^c^	2.28 (1.40–3.73) **	2.88 (1.71–4.85) **	2.35 (1.35–4.08) **	3.14 (1.72–5.74) ***	1.59 (0.76–3.32)	1.72 (0.79–3.73)
Stroke	0.89 (0.57–1.40)	0.98 (0.62–1.56)	0.90 (0.54–1.50)	0.83 (0.49–1.42)	1.11 (0.63–1.98)	1.23 (0.67–2.23)
Myocardial infarction	0.45 (0.11–1.80)	0.42 (0.10–1.71)	0.57 (0.14–2.33)	0.75 (0.18–3.16)	0.39 (0.06–2.83)	0.36 (0.05–2.68)
Cognitive Impairment ^d^	1.50 (0.92–2.42)	1.56 (0.95–2.56)	1.85 (1.02–3.32) *	2.22 (1.18–4.16) *	1.21 (0.66–2.20)	1.30 (0.70–2.41)
Nutritional Vulnerability ^e^	1.10 (0.77–1.56)	1.20 (0.82–1.75)	1.17 (0.79–1.76)	1.38 (0.89–2.12)	0.87(0.54–1.43)	0.87 (0.52–1.47)
Falls in the past 6 months ^f^	2.18 (1.48–3.20) ***	2.22 (1.47–3.34) ***	2.03 (1.34–3.08) ***	1.93 (1.23–3.01) **	1.65 (0.99–2.73)	1.65 (0.98–2.79)
Hospitalisation for ≥1 night	–	–	2.99 (2.00–4.47) ***	2.96 (1.93–4.55) ***	–	–

* <0.05, ** <0.01, *** <0.001; ^a^ Poor composite outcome: death and/or hospital transfer during the study period; ^b^ CFS = Clinical Frailty Scale; ^c^ COPD = Chronic obstructive pulmonary disease; HR = hazard ratio; ^d^ Cognitive impairment = at least one of: moderate cognitive impairment or dementia, long-tem mental disorder, acute behavioural alterations (depression, psychosis); ^e^ Nutritional vulnerability criteria = at least two of weight loss, diet modification, malnutrition or presence of feeding tube. ^f^ Falls = More than 2 falls in the past 6 months.

**Table 3 healthcare-08-00284-t003:** Adjusted associations between predictors and poor composite outcome. Final Cox regression models for 3 different outcomes.

Adjusted Predictors of Poor Follow-Up Outcome	Poor Composite Outcome ^§^		Death		Hospital Admission	
	HR (95% CI)	*p*-Value	HR (95% CI)	*p*-Value	HR (95% CI)	*p*-Value
Age	1.02 (0.99–1.05)	0.192	1.03 (1.00–1.06)	0.071	1.01 (0.98–1.05)	0.568
Male	1.41 (0.97–2.05)	0.070	1.11 (0.72–2.73)	0.628	1.78 (1.09–2.91)	**0.022**
COPD	2.64 (1.59–4.39)	**<0.001**	3.22 (1.78–5.81)	**0.0001**		
Cognitive impairment	1.49 (0.91–2.42)	0.111	2.22 (1.18–4.15)	**0.013**		
Falls in past 6 months	2.14 (1.46–3.12)	**<0.001**	2.02 (1.31–3.14)	**0.002**	1.64 (0.98–2.75)	0.059
Hospital admission			2.99 (1.97–4.53)	**<0.0001**		

^§^ Poor Composite Outcome: death and/or hospital admission; N/S = not significant; COPD = chronic obstructive pulmonary disease; HR = Hazard Ratio; CI = Confidence Interval; **Bolded** numbers indicate statistical significance.

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
