# Peer review of "Feasibility of Using a Risk Assessment Tool to Predict Hospital Transfers or Death for Older People in Australian Residential Aged Care. A Retrospective Cohort Study"

_healthcare, 2020, doi:10.3390/healthcare8030284_

Round 1

Reviewer 1 Report

Dear authors,

Thank you for this interesting work.

I have made some remarks

Your abstract needs clarification. Please give the proper background, a clear study aim and method and the conclusion of your results in the abstract.

274 the modified CrisTAL. You state that you added 3 variables but you describe 4: polypharmacy, falls, pneumonia and nutritional state. I presume you excluded polypharmacy but make this clear.

285-287 refrase the aim. The sentence is very difficult to understand. The aim is to study the effect of the CristAL tool on the identification of patients who benefit of EOL discussions.

289 what are the last months of life? 6-12?

328-332 these outcomes are the objectives of your study. Bring this forward to aims/objectives

344 p<0.15 makes your research statiscially more weak

349 what definition of dementia do you use?

368 internationally the ‘very old’ are people of 85 years or older.

372 it is better to describe here your definition of a frail person and the cutoff. Line 447

379 make very clear what is the poor composite outcome: hospital admission and/or death? Is the hospital admission related to the disease of the patient? What exactly is hospital admission? A consult or admission to a ward? Also short-stay?

410 the data that are not collected for some reason should be in limitations

Discussion

What does your work add to the topic of nursing home resident hospital transfers? How do your results relate to known literature?

425-427 and 432-434 Don’t confuse the discussion with limitations.

440 you should distinguish between a fall requiring hospital admission and adverse outcome. Please include references that support your theory.

Author Response

REVIEWER 1

Thank you for this interesting work.

I have made some remarks

Your abstract needs clarification. Please give the proper background, a clear study aim and method and the conclusion of your results in the abstract.

AUTHORS REPLY: The MDPI abstracts do not have subheadings and the word limit is very strict . We believed we had covered all of these points. However, we have amended the abstract to incorporate the reviewer’s recommended changes, which make it more obvious to readers what the aims, methods and conclusions are. Page 1.

274 the modified CrisTAL. You state that you added 3 variables but you describe 4: polypharmacy, falls, pneumonia and nutritional state. I presume you excluded polypharmacy but make this clear.

AUTHORS REPLY: Thanks for picking this up. We meant 4 variables and already state later that we could not measure them all. Line 276, page2.

285-287 refrase the aim. The sentence is very difficult to understand. The aim is to study the effect of the CristAL tool on the identification of patients who benefit of EOL discussions.

AUTHORS REPLY: We have modified the aim substantially with wording similar to that suggested by the reviewer. We are not measuring an effect but the feasibility of using the tool. Lines 288-291 page 2.

289 what are the last months of life? 6-12?

AUTHORS REPLY: Yes, for the purpose of this study we have clarified that it is the last 6-12 months in the introduction and objectives. Lines 280 and  293, page 2.

328-332 these outcomes are the objectives of your study. Bring this forward to aims/objectives

AUTHORS REPLY: Respectfully, we do not agree that the outcomes ARE the objectives of the study. However, we have brought them forward closer to the objectives, so readers can more eaily establish the link. Lines 299-304., pages 2 & 3.

344 p<0.15 makes your research statiscially more weak

AUTHORS REPLY: We agree, in principle. However, the point of this analysis was to find not only statistically significant associations but also clinically meaningful ones. Setting a less stringent cut-point is common in the literature for this purpose. One of the articles (Heinze 2017) explains it better than we can: “Eliminating weak effects is dangerous as in etiologic studies, bias could result from falsely omitting an important confounder. This is because regression coefficients generally depend on which other variables are in a model, and consequently, they change their value if one of the other variables are omitted from a model”. We have added a brief justification for this choice and a reference to support it. Lines 363-364 on page 4 and  new reference #24.

349 what definition of dementia do you use?

AUTHORS REPLY: Staff used a combination of sources to supplement the PAS. We have given a reference and added a small paragraph presenting this as an example of how data was documented.  Lines 323-328 on page 3 and new reference #23.

368 internationally the ‘very old’ are people of 85 years or older.

AUTHORS REPLY: The median age in our subjects was 84 years, but we agree and have deleted the word ‘very’ form the context. Line 388, page 5.

372 it is better to describe here your definition of a frail person and the cutoff. Line 447

AUTHORS REPLY: We have moved the name of the scale, the reference and added the cut-off to the early paragraphs of the results. Line 390 on page 5 and reference #25.

379 make very clear what is the poor composite outcome: hospital admission and/or death? Is the hospital admission related to the disease of the patient? What exactly is hospital admission? A consult or admission to a ward? Also short-stay?

AUTHORS REPLY: We have clarified composite outcome and inserted a definition of hospital admission in the methods, under ‘data collection’. Lines 297 p2, 417 p7 & 427 p8,  333-335 on page 3.

410 the data that are not collected for some reason should be in limitations

AUTHORS REPLY: We have moved the comment on lack of information on sameday returns from hospital to the ‘limitations’ section. Lines 510-513 on page 10.

Discussion

What does your work add to the topic of nursing home resident hospital transfers?

We have emphasised the take home message in the discussion now.  Lines 432-435 on page 8.

How do your results relate to known literature?

AUTHORS REPLY: We believe we had already addressed this under the section 4.1 “comparison with other studies”. Lines 463-489 on page 9.

425-427 and 432-434 Don’t confuse the discussion with limitations.

AUTHORS REPLY: We have also moved this paragraph to the limitations section (lines 498-503 on page 9.

440 you should distinguish between a fall requiring hospital admission and adverse outcome. Please include references that support your theory.

AUTHORS REPLY: We have added a caveat that we cannot ascertain falls requiring hospitalisation or not, and added a reference for falls being an adverse event. Lines 468-469 on page9 and new reference #30.

Reviewer 2 Report

Specify research methodology, including data that has been collected and data that could not be collected in the databases.

In the discussion, clarify whether other tools would have been more relevant to data analysis because of the many missing data.

Due to several crucial missing data (e.g. nutrition, polypharmacy) to be able to draw conclusions from the study in the study population, the findings should be interpreted with caution.

Author Response

REVIEWER 2

Specify research methodology, including data that has been collected and data that could not be collected in the databases.

AUTHORS REPLY: we believe have already done this under the relevant sections of the original manuscript (methods section 2.2 on page 3, and limitations of the study section 4.2 on pages 9-10) but have enhanced their visibility in the new version.

In the discussion, clarify whether other tools would have been more relevant to data analysis because of the many missing data.

AUTHORS REPLY: We wish we could recommend another tool but from our research and reading of the recent literature we believe this is the first attempt to  prognosticate risk of transfer or death in residential aged care. Many of the published studies are descriptive and focus on the main causes of hospital presentation. The few we found mentioning potentially avoidable transfers refer to ambulatory-care sensitive conditions. Under ‘comparison with other studies’ we have added a statement about this. Lines  481-485 om page 9, and new references 33 & 34.

Due to several crucial missing data (e.g. nutrition, polypharmacy) to be able to draw conclusions from the study in the study population, the findings should be interpreted with caution.

AUTHORS REPLY:  We  were able to document nutritional vulnerability (Table 1) but have noted the problem with the low prevalence of malnutrition and inability to document polypharmacy under the ‘implications of the study’. Lines 531-532 on page 10.

Reviewer 3 Report

Authors clearly indicate the limitations of this study. Since about half of the participants were not native to Australia it would seem of significance particularly if you are writing about advance care planning. I find the study interesting in its focus on advance directives when it seems as if approach to care would of benefit to these participants. It seems you are trying to find data to recommend advance care planning which I commend. However, the complexity of the official planning should not be overlooked.

Author Response

REVIEWER 3

Authors clearly indicate the limitations of this study. Since about half of the participants were not native to Australia it would seem of significance particularly if you are writing about advance care planning. I find the study interesting in its focus on advance directives when it seems as if approach to care would of benefit to these participants. It seems you are trying to find data to recommend advance care planning which I commend. However, the complexity of the official planning should not be overlooked.

AUTHORS REPLY: Thanks for the comment. Accordingly, we have added a statement about future research under the ‘comparison with other studies’ section. Lines 489-491 on page 9.